# AUXILIARY TASK UPDATE DECOMPOSITION: THE GOOD, THE BAD AND THE NEUTRAL

**Lucio M. Dery**
Department of Computer Science
Carnegie Mellon University
Pittsburgh, PA, USA

**Yann Dauphin**
Google Research

**David Grangier**
Google Research

## ABSTRACT

While deep learning has been very beneficial in data-rich settings, tasks with smaller training set often resort to pre-training or multitask learning to leverage data from other tasks. In this case, careful consideration is needed to select tasks and model parameterizations such that updates from the auxiliary tasks actually help the primary task. We seek to alleviate this burden by formulating a model-agnostic framework that performs fine-grained manipulation of the auxiliary task gradients. We propose to decompose auxiliary updates into directions which help, damage or leave the primary task loss unchanged. This allows weighting the update directions differently depending on their impact on the problem of interest. We present a novel and efficient algorithm for that purpose and show its advantage in practice. Our method leverages efficient automatic differentiation procedures and randomized singular value decomposition for scalability. We show that our framework is generic and encompasses some prior work as particular cases. Our approach consistently outperforms strong and widely used baselines when leveraging out-of-distribution data for Text and Image classification tasks.

## 1 INTRODUCTION

Multitask learning (Caruana, 1997) and pretraining (Devlin et al., 2018; Caron et al., 2019) have transformed machine learning by allowing downstream tasks with small training sets to benefit from statistical regularities from data-rich related tasks (Collobert & Weston, 2008; Zhang et al., 2014; Liu et al., 2019; Kornblith et al., 2019). Despite these advances, leveraging the mixing of tasks is still an art left to the practitioner. When one is interested in a *primary* task, it is unclear how to select helpful auxiliary tasks, an appropriate parameter sharing architecture and a good way to filter out auxiliary data which might be detrimental to the primary tasks. Without careful choices, pre-training might hurt end-task performance (Gururangan et al., 2020) or have limited impact (Raghu et al., 2019).

Prior work has examined these problems and proposed solutions, either to choose auxiliary tasks depending on their impact on the primary task (Du et al., 2018; Lin et al., 2019) or to equalize the impact of updates across tasks (Sener & Koltun, 2018; Chen et al., 2018; Hessel et al., 2019). Recently, several approaches (Sinha et al., 2018; Suteu & Guo, 2019; Yu et al., 2020) have been proposed that attempt to minimize interference between the updates across tasks. Our work builds on this direction, but unlike these previous approaches, we do not consider a symmetric view of multi-task learning in the sense that our goal is not to train a model performing well on all tasks. Instead, we focus on improving generalization for a single task, the primary task, and the other tasks, the auxiliary tasks are considered only through their impact on the problem of interest.

For that purpose, we introduce a framework which decomposes the gradient updates from the auxiliary tasks according to their impact on the primary task. We analyze the auxiliary task gradients in the subspace spanned by the primary task per-example gradients. This allows us to decompose auxiliary gradients into into three components : components that help, interfere or have no impact on the primary task according to the Taylor expansion of the expected primary loss. This decomposition allows us to re-weight each component differently prior to the update. Our framework enables us to treat each auxiliary update differently depending on its impact on the task of interest and it

encompasses prior methods such as classical multitask learning (Caruana, 1997) or more novel gradient surgery techniques (Yu et al., 2020). To achieve a tractable approach, we introduce an efficient, robust algorithm (ATTITTUD, Auxiliary Task Training with Influence from Target Task Update Direction) to estimate the subspace spanned by the primary task gradients in an online manner and decompose the auxiliary updates appropriately. As a result, we can integrate our approach with the stochastic training of large neural networks in various contexts.

The contribution of our work is four-fold. To our knowledge, this paper proposes the first approach to adapt auxiliary gradients using a decomposition built from the span of the primary task Jacobian. In order to scale this approach to deep neural nets, we contribute a tractable and efficient algorithm called ATTITTUD that leverages insights from randomized linear algebra and automatic differentiation such as the R-operator (Pearlmutter, 1994). As our third contribution, we show that the fine-grained manipulation of the auxiliary task gradients under ATTITTUD, represents a unified framework that encompasses several previous approaches to asymmetrical task learning as special cases. Finally, we demonstrate the efficacy of our approach in both data-rich and data-starved primary tasks, over both images and textual data.

## 2   RELATED WORK

Methods to leverage data outside of the task of interest have been popular in machine learning since the inception of multitask learning (Caruana, 1997; Ruder, 2017; Vandenhende et al., 2020). These methods address multiple task simultaneously and have been successful in various application domains (Collobert & Weston, 2008; Zhang et al., 2014; Misra et al., 2016). The optimization problem induced by multitask learning is difficult and solutions have been proposed for the various difficulties, including dealing with task gradients of different magnitude (Sener & Koltun, 2018; Chen et al., 2018; Hessel et al., 2019), or interfering with each others (Sinha et al., 2018; Suteu & Guo, 2019; Yu et al., 2020). The specific problem of interference has been studied extensively in the context of continual learning. Continual learning visits task in sequence and update interference is particularly problematic as it yields newer tasks to damage previously mastered tasks. In particular, a family of methods to project the gradient of the new tasks to be orthogonal to the gradient of the previous tasks has been proposed (Lopez-Paz & Ranzato, 2017; Chaudhry et al., 2018; Farajtabar et al., 2019).

Different from many previous approaches, we are not interested in addressing multiple tasks per se. In our setting, only the *primary* task matters and the other *auxiliary* task have the sole role of improving generalization on the primary task. This is the setting considered by Du et al. (2018); Lin et al. (2019), who favor auxiliary tasks whose gradient directions are helpful to the primary task. Unlike these works that use coarse properties like the cosine similarity between averaged gradients, our approach allows fine-grained gradient manipulation within a subspace. Also, in our case, we do not distinguish between the different auxiliary tasks. Instead, we aim at correcting every auxiliary gradient in the same manner to improve the loss on the primary task. This type of gradient correction is related to Yu et al. (2020), which considers projecting multi-task gradients such that the directions of disagreement are removed. This method is actually a special case of our framework.

Our work also shares some similarities with data selection and domain adaptation approaches. In this case, the training data comes from a single task but its distribution is different from the validation/test distribution (Moore & Lewis, 2010; Axelrod et al., 2011; Ngiam et al., 2018). This classical problem has recently been addressed by sampling training points whose gradient aligns well with the expected validation gradient (Wang et al., 2020b;a). Instead of sampling individual points based on an estimated distribution of how helpful they will be to the primary task, our work avoids the use (and inherent challenges) of this reinforcement learning approach by operating on batch gradients of groups of points.

Our primary task/auxiliary task setting is also related to the pre-training then fine-tuning paradigm in which the auxiliary tasks are visited first (pre-training) to give an initialization for training on the primary task (fine-tuning). These methods have been very successful in settings where primary task data are rare. In particular, it is common to first rely on an unsupervised task over very large datasets prior to fine tuning over a supervised task (Devlin et al., 2018; Liu et al., 2019; Kornblith et al., 2019; Yang et al., 2019; Song et al., 2019; Caron et al., 2018).

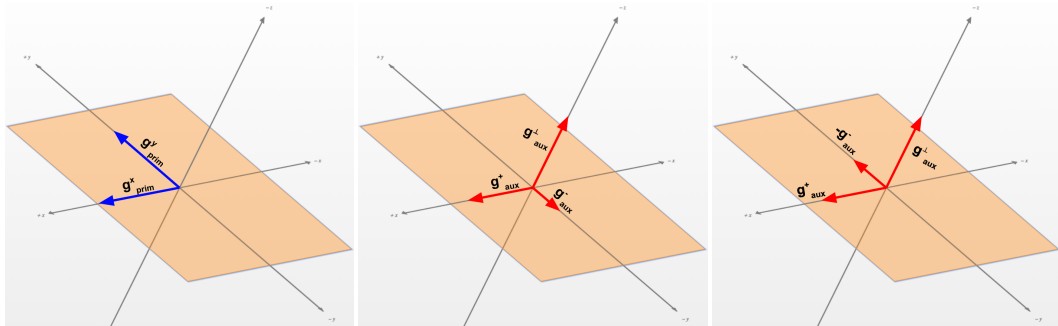

Figure 1: Example gradient manipulation in the 2-D $x - y$ plane with ATTITUD. ATTITUD can operate in any n-dimensional subspace. **Left**: Primary task gradient $g_{prim}$ decomposed along the 3 Dimensions $x$, $y$ and $z$. **Mid**: Decomposed Auxiliary task gradient $g_{aux}$. We label the $x$ component of $g_{aux}$ as positive since it agrees (in direction) with the $x$ component of $g_{prim}$. Since the $y$ component of $g_{aux}$ is in the opposite direction as that of $g_{prim}$, this is assigned a negative label. **Right**: Corresponds to $\tilde{g}_{aux}$ obtained by applying $\eta_{aux} = (1.0, 1.0, -1.0)$. We flip the conflicting gradient direction to agree with our primary task. This is just one configuration achievable under our framework.

## 3 AUXILIARY TASK UPDATE DECOMPOSITION

This section introduces a new method to improve generalization on a primary task $T^*$ using training data from auxiliary tasks $\mathbb{T} = \{T_1, \ldots, T_n\}$, where $\theta \in \mathbb{R}^D$ denote the parameters shared by all tasks. Our approach leverages gradient updates from the auxiliary tasks, but unlike the traditional approach, we decompose these gradients to maximize their usefulness to $T^*$. Precisely, we decompose the auxiliary task gradients into directions which decrease a first-order approximation of the primary task loss, increase it or have no effect. This decomposition allows weighting these three directions differently when learning from the auxiliary tasks.

In order to decompose the auxiliary gradient, we must collect more fine-grained statistics about the primary task. At each training step, we collect the gradient of the loss with respect to $\theta$ for individual examples from the primary task, $\{\nabla_\theta \mathcal{L}_i^{\text{prim}}, \forall i \}$. The span of these vectors,

$$\mathcal{S} = \text{Span}\{\nabla_\theta \mathcal{L}_i^{\text{prim}}, \forall i\}$$

defines a subspace in which any linear combination of primary task gradients lies, including the gradient of the expected primary task loss, i.e. $g_{prim} = \mathbb{E}(\nabla_\theta \mathcal{L}_i^{\text{prim}}) \in \mathcal{S}$. We denote the size of the subspace, $|\mathcal{S}| = K$. This is upper-bounded by the number of examples $m$, used to construct $\mathcal{S}$. If we define the orthogonal complement of $\mathcal{S}$ as $\mathcal{S}^\perp$, any vector $v \in \mathcal{S}^\perp$, is therefore orthogonal to $g_{prim}$, i.e. $v \cdot g_{prim} = 0$. This means that adding such a vector to the parameters has no impact on the expected primary task loss, according the order-1 Taylor expansion of $\mathcal{L}^{\text{prim}}$, i.e.

$$\mathcal{L}^{\text{prim}}(\theta + v) \simeq \mathcal{L}^{\text{prim}}(\theta) + v \cdot g_{prim} = \mathcal{L}^{\text{prim}}(\theta).$$

We propose to project auxiliary task gradients onto $\mathcal{S}$ and $\mathcal{S}^\perp$. This allow to distinguish between the directions of the auxiliary task updates which impact the primary task loss and those which do not. If we denote the averaged auxiliary task gradient as $g_{aux} = \mathbb{E}(\nabla_\theta \mathcal{L}_i^{\text{aux}})$, we can decompose this gradient as $g_{aux} = g_{aux}^\pm + g_{aux}^\perp$. where $g_{aux}^\pm \in \mathcal{S}$ is the portion of the gradient that lies in the span of the primary task example gradients and $g_{aux}^\perp \in \mathcal{S}^\perp$ is the portion that lies outside of it. Since $g_{aux}^\perp \in \mathcal{S}^\perp$, it is orthogonal to the average primary task gradient and parameter updates along the direction of $g_{aux}^\perp$ are expected to have limited impact on the primary task loss. On the other hand, updates along the direction of $g_{aux}^\pm$ can potentially improve or damage the averaged primary task loss. This component deserves a more careful treatment.

For that purpose, we introduce $\{u_i, i = 1, \ldots, K\}$ an orthonormal basis of $\mathcal{S}$. In this basis, we can measure if the components of $g_{aux}^\pm$ agree or disagree with $g_{prim}$. We say that the two gradients agree along $u_i$ iif $\text{sign}(g_{aux}^\pm \cdot u_i) = \text{sign}(g_{prim} \cdot u_i)$. This means that we can decompose $g_{aux}^\pm = g_{aux}^+ + g_{aux}^-$

where $g_{aux}^+$ refers to the projection of $g_{aux}^\pm$ onto the basis vectors where $g_{aux}^\pm$ and $g_{prim}$ agree. By this decomposition, $g_{aux}^+$ helps the primary task, $g_{aux}^+ \cdot g_{prim} > 0$, while $g_{aux}^-$ interfere with the primary task, $g_{aux}^- \cdot g_{prim} < 0$.

Guided by the primary task, we can therefore decompose the auxiliary task gradient as

$$g_{aux} = g_{aux}^\perp + g_{aux}^+ + g_{aux}^-$$  (1)

which is described on Fig 1. Our approach proposes to re-weight differently the components of $g_{aux}$, i.e.

$$\tilde{g}_{aux} = \eta_\perp g_{aux}^\perp + \eta_+ g_{aux}^+ + \eta_- g_{aux}^-$$  (2)

where $\boldsymbol{\eta}_{aux} = (\eta_\perp, \eta_+, \eta_-)$ are hyper-parameters adjusting the auxiliary gradient according to the impact on the main task. If we also wish to include the primary task gradient in descent, as with multitasking, we can introduce $\boldsymbol{\eta}_{prim}$ as a scalar control variable to modulate its weighting.

A consequence of introducing $\boldsymbol{\eta}_{aux}$ is that specific configurations lead us to gradient updates that are guaranteed to do no harm to both tasks. This is captured by Theorem 1 below.

**Theorem 1.** *Let $\mathcal{L}^{\mathrm{aux}}(\theta_t)$ and $\mathcal{L}^{\mathrm{prim}}(\theta_t)$ represent the full batch losses of the auxiliary tasks and primary task respectively at step t. We assume the gradients of $\mathcal{L}^{\mathrm{aux}}$ and $\mathcal{L}^{\mathrm{prim}}$ are Lipschitz continuous with constant $L > 0$. Following the update rule : $\theta_{t+1} = \theta_t - \alpha \cdot \tilde{g}_{aux}$, where $\alpha \leq \frac{1}{L}$ is the learning rate, we are guaranteed :*

$$\mathcal{L}^{\mathrm{aux}}(\theta_{t+1}) \leq \mathcal{L}^{\mathrm{aux}}(\theta_t)$$
$$\mathcal{L}^{\mathrm{prim}}(\theta_{t+1}) \leq \mathcal{L}^{\mathrm{prim}}(\theta_t)$$

*If $\eta_- = 0$ and $\eta_\perp, \eta_+ \geq 0$*

*Proof.* See Appendix A ☐

This theorem focuses on a single update and guarantees progress on both auxiliary and primary tasks. However, our asymmetric scenario is not interested in improving the auxiliary tasks per se and is amenable to more aggressive settings. Ideally we want gradient updates during pre-training with $\mathbb{T}$ to not only do-no-harm to $T^*$ when applied downstream but also to be along descent directions that are maximally beneficial to $T^*$. We can consider $\eta_- < 0$ as in Fig 1. Reversing the direction of $g_{aux}^-$ by setting $\eta_- < 0$ preserves the descent guarantee on $\mathcal{L}_{prim}(\theta_{t+1})$ but no longer ensures descent on $\mathcal{L}_{aux}(\theta_{t+1})$. There are other interesting settings for our control parameters. One can recover the original gradient $g_{aux}$ with $\eta_\perp = \eta_- = \eta_+ = 1.0$. One can choose to drop gradients orthogonal to the primary task gradient span with $\eta_\perp = 0.0$, or ignore those which conflict with the main task by setting $\eta_- = 0.0$.

**Relationships to other approaches**  Our framework is generic and encompasses other approaches as a particular case. One can train solely on the primary task by selecting $\boldsymbol{\eta}_{aux} = (0.0, 0.0, 0.0)$ and $\boldsymbol{\eta}_{prim} = 1.0$. Classical multitasking corresponds to $\boldsymbol{\eta}_{aux} = (1.0, 1.0, 1.0)$ and $\boldsymbol{\eta}_{prim} > 0.0$, while classical pre-training corresponds to performing a first phase with $\boldsymbol{\eta}_{aux} = (1.0, 1.0, 1.0)$ and $\boldsymbol{\eta}_{prim} = 0.0$. Interestingly, our formulation introduces novel variants of pre-training, for instance, one can consider pre-training with only auxiliary gradients helpful to the primary task, $\boldsymbol{\eta}_{aux} = (0.0, 1.0, 0.0)$ and $\boldsymbol{\eta}_{prim} = 0.0$, followed by fine-tuning with $\boldsymbol{\eta}_{aux} = (0.0, 0.0, 0.0)$ and $\boldsymbol{\eta}_{prim} = 1.0$.

Our approach also instantiates PCGrad (Yu et al., 2020) as a particular case. This method was introduced to address the issue of conflicting gradients in multitask settings. PCGrad orthogonalizes the gradients of each task and removes conflicting gradients. To recover PCGrad under our approach, note that it is equivalent to a specific choice of our decomposition in the 1-D subspace spanned by the $g_{prim}$. PCGrad then removes components of $g_{aux}$ that conflict with $g_{prim}$ which is equivalent to $\boldsymbol{\eta}_{aux} = (\alpha_{aux}, \alpha_{aux}, 0.0)$ and $\boldsymbol{\eta}_{aux} = \alpha_{prim}$.

## 4 IMPLEMENTATION

Equation 2 requires selecting a basis for the span of primary task gradients. Multiple choices are possible to define the basis $\{u_i\}$, to represent the span at each optimization time-step. This choice

is important since the components of $\boldsymbol{g}_{aux}^{\pm}$ are labeled positive or negative depending on how they agree with the projection of the averaged primary task gradient onto the same basis. A natural choice is to select the basis as the singular vectors of the matrix of primary task per-example gradients $\boldsymbol{J}^* \in \mathbb{R}^{m \times D}$, also know as the Jacobian. To improve efficiency and prevent over-fitting on a few examples, we consider the span defined by the, $k < |\mathcal{S}|$, largest principal vectors of $\boldsymbol{J}^*$. Using the principal vectors as directions of descent instead of the mean induces a more robust algorithm since the mini-batch average gradient is susceptible to outliers and skew from replicated data-points. To the best of our knowledge, we are the first to propose using the singular vectors of $\boldsymbol{J}^*$ as directions of descent. We leave the theoretical implications of this algorithm to future work but note that its variance reduction properties may induce generalization benefits (Namkoong & Duchi, 2017).

We also consider alternative choices of bases as baselines, including the canonical parameter basis. This choice will examine the sign of every parameter update to verify whether it agrees with $\boldsymbol{g}_{prim}$. Whilst Theorem 1 holds irrespective of the choice of basis, its proof reveals that the amount of progress made on each loss depends on the choice of basis. Specifically, the reduction in $\mathcal{L}^{\mathrm{prim}}(\theta_{t+1}), \mathcal{L}^{\mathrm{aux}}(\theta_{t+1})$ after a gradient step along $\tilde{\boldsymbol{g}}_{aux}$ is proportional to the fraction of the norms of $\boldsymbol{g}_{prim}$ and $\boldsymbol{g}_{aux}$ captured by the subspace spanned by our choice of basis. To justify our use of the top singular values of $\boldsymbol{J}^*$, we evaluate this fraction for different choice of basis in our experiments (see Appendix C).

We are interested in applying our approach to the training of large neural networks and must consider a scalable algorithmic solution. As stochastic optimization is prevalent in this setting, we construct subspace S from a mini-batch of primary task data. Similarly, the expected gradients $\boldsymbol{g}_{prim}$ and $\boldsymbol{g}_{aux}$ are defined over a mini-batch. Instead of computing the singular value decomposition (SVD) of $\{\nabla_\theta \mathcal{L}_i^{\mathrm{prim}}, \forall i\}$ exactly, we rely on a randomized approximation (Halko et al., 2011; Rokhlin et al., 2010; Nakatsukasa, 2017). This method does not require instantiating the vectors $\{\nabla_\theta \mathcal{L}_i^{\mathrm{prim}}, \forall i\}$ and only needs a low dimensional projection onto a random subspace. This is advantageous for high dimensional cases, i.e. when the number of model parameters is large. In our case, this method also allows us to benefit from memory-efficient computation of Jacobian Vector product using the R-operator (Pearlmutter, 1994) offered by automatic differentiation packages (Baydin et al., 2015) like Pytorch (Paszke et al., 2017). This means that we can compute SVD with a limited computational and memory burden, albeit without sacrificing approximation accuracy (Nakatsukasa, 2017). Additionally, we do not recompute the basis at every optimization step but at every $n$ steps, which is efficient when training with small updates, e.g. when small learning rates and gradient clipping are used (Pascanu et al., 2013) (see Appendix C for more details about $n$).

We study the impact of these choices in practice in Section 6. Putting it all together results in the ATTITTUD algorithm, Auxiliary Task Training with Influence from Target Task Update Direction, shown as Algorithm 1. The sub-procedure `randomized_lowrank_approx` is detailed in Appendix B as Algorithm 2 .

---

**Algorithm 1:** ATTITTUD : Construct Auxiliary Task Surrogate Gradient

---

**Require :** $\boldsymbol{g}_{aux}, \boldsymbol{J}^*$ : Auxiliary task average gradient, primary task Jacobian
**Require :** $\boldsymbol{\eta}_{aux} = (\eta_\perp, \eta_+, \eta_-)$ : Auxiliary task control parameters
**Require :** $k$ : Size of subspace

$\boldsymbol{g}_{prim} = \frac{1}{m} \sum_{i=1}^{m} \boldsymbol{J}_{i,:}^*$

$\boldsymbol{V} \leftarrow$ `randomized_lowrank_approx`$(\boldsymbol{J}^*, k)$

$\boldsymbol{p}_{prim}, \boldsymbol{p}_{aux} = \boldsymbol{V}_t(\boldsymbol{g}_{prim})^T, \boldsymbol{V}_t(\boldsymbol{g}_{aux})^T$

// ∘ is the hadamard product operator

$\boldsymbol{p}_{aux}^+, \boldsymbol{p}_{aux}^- = \left( \mathbf{1}_{\left[ \mathrm{p}_{prim} \circ \mathrm{p}_{aux} \geq 0 \right]} \right) \circ \boldsymbol{p}_{aux}, \left( \mathbf{1}_{\left[ \mathrm{p}_{prim} \circ \mathrm{p}_{aux} < 0 \right]} \right) \circ \boldsymbol{p}_{aux}$

// Calculate the decomposition components

$\boldsymbol{g}_{aux}^+, \boldsymbol{g}_{aux}^- = \left( \boldsymbol{p}_{aux}^+ \right)^T \boldsymbol{V}, \left( \boldsymbol{p}_{aux}^- \right)^T \boldsymbol{V}$

// Calculate the out of span component

$\boldsymbol{g}_{aux}^\perp = \boldsymbol{g}_{aux} - \left( \boldsymbol{g}_{aux}^+ + \boldsymbol{g}_{aux}^- \right)$

$\tilde{\boldsymbol{g}}_{aux} = \left( \eta_\perp \cdot \boldsymbol{g}_{aux}^\perp \right) + \left( \eta_+ \cdot \boldsymbol{g}_{aux}^+ \right) + \left( \eta_- \cdot \boldsymbol{g}_{aux}^- \right)$

**Return :** $\tilde{\boldsymbol{g}}_{aux}$ : Auxiliary task surrogate gradient

---

## 5 EXPERIMENTAL SETUP

We compare ATTITTUD with previous methods on a variety of tasks and domains. We rely on both text and image classification tasks to conduct our analysis. We also present ablation experiments to explain the impact of hyper-parameter selection. We make code for ATTITTUD and related experiments available on github. [1]

**Text Classification.** We apply our method on binary sentiment classification. We consider the Amazon Helpfulness (McAuley et al., 2015) and Imdb Movie Review (Maas et al., 2011) tasks. The Amazon Helpfulness task splits text reviews into 115k/5k/25k documents for train-validation-test split whilst the Imdb Review dataset has a 20k/5k/25k split. The Imdb Review task also has 50k unlabeled reviews as extra data which we utilize.

For our models we build on top of Gururangan et al. (2020)'s work where they introduce Task-Adaptive Pre-training (TAPT). TAPT further pre-trains a generic model, Roberta (Liu et al., 2019), by performing Masked Language Modelling, MLM, (Devlin et al., 2018) on the task specific data (ignoring the labels) before doing supervised learning with the same data. We replicate Gururangan et al. (2020)'s experimental setup and re-use their hyper-parameters for our experiments. We use the TAPT task as our auxiliary task. We extend TAPT to use our method by modifying the TAPT gradient with guidance from the supervised-learning task gradients. As baselines, we compare against TAPT and cross-TAPT: where we swap the masked language modelling pre-training data for the two tasks. Cross-TAPT is a setting where one uses out-of-distribution data for pre-training.

**Image Classification.** We apply our method to both high-resource and limited-data image classification tasks. We use the Cifar100 dataset (Krizhevsky et al., 2009) to explore the high-resource setting. We follow Rosenbaum et al. (2017) and treat each of the 20 super-classes / coarse labels of Cifar100 as a separate task. In our asymmetrical task setting, each of the 20 tasks is treated as a primary task, whilst the remaining 95 classes are grouped into a single auxiliary task. Thus, for each coarse label, we have an auxiliary 95-way classification task and a 5-way primary classification task. Moving forward, we refer to this setting as MultiCifar100.

We use a down-sampled version of Cifar10 (Krizhevsky et al., 2009) as a low-resource setting. Specifically, we rely on Cat-vs-Dog for the primary task and use the remaining 8 classes for the auxiliary task. Our auxiliary task is therefore an 8-way classification task where each class has 5,000 examples. We restrict the Cat and Dog classes to only 50 training examples from each class. We use the low-resource setting to compare against other methods and for our ablation study.

For these vision experiments, we use a WideResNet-22 architecture (Zagoruyko & Komodakis, 2016) with a depth of $k = 4$. We compare our method to 4 different baselines : no pre-training, vanilla pre-training, multitasking and PCGrad (Yu et al., 2020). Our architecture is more standard and allows gradient descent optimization unlike the routing network of Rosenbaum et al. (2017) and (Yu et al., 2020), which requires reinforcement learning for training.

**Medical Imaging Transfer.** We apply our method to cross-domain transfer for low-resource medical image classification. Specifically, we use 5k training examples from the ChexPert Dataset (Irvin et al., 2019) as our primary task and seek to identify 5 different thoracic pathologies: atelectasis, cardiomegaly, consolidation, edema and pleural effusion. This setup has been used in several cross-domain pretraining studies (Raghu et al., 2019; Jaiswal et al., 2019). Note that since we do not have access to the test set for this task, we use the validation set (231 images) as a proxy test set, and sample 100 images from the training data as a new validation set. We rely on generic photographs (Imagenet) as an auxiliary task (Deng et al., 2009). We use Tiny Imagenet Dataset (Le & Yang, 2015), a subset of Imagenet which consists of 500 examples each from 200 classes, instead of training on full Imagenet. All approaches are applied to the Resnet18 model (He et al., 2016) trained with Adam (Kingma & Ba, 2014).

For all our experiments, we select the auxiliary task control parameters $\boldsymbol{\eta}_{aux}$ within $\{(1.0, 1.0, -1.0), (1.0, 1.0, 0.0), (1.0, 0.0, -1.0), (1.0, 0.0, 0.0)\}$ for ease of interpretability. For settings where we compare against multi-tasking, we select $\boldsymbol{\eta}_{prim}$ within a small subset of the settings that worked best with multitasking baseline experiments. These choices limit the overhead of hyper-parameter search but still allow us to show the empirical advantage of our method. In

---

[1]Code available here https://github.com/ldery/ATTITTUD

| | Imdb | Imdb + Amazon MLM | Amazon | Amazon + Imdb MLM |
|---|---|---|---|---|
| Roberta | $95.4 \pm 0.14$ | - | $67.0 \pm 0.50$ | - |
| TAPT | $96.1 \pm 0.11$ | $95.1 \pm 0.10$ | $70.3 \pm 0.87$ | $67.8 \pm 0.46$ |
| Ours | $96.1 \pm 0.09$ | **95.4±0.03** | $70.1 \pm 1.13$ | **68.5±1.01** |

Table 1: Results on Text Classification measured by F1. Experiments are averaged over 5 runs.

all our experiments, we provide all methods with similar hyper-parameter search budgets, e.g. for Cifar10-Cat-vs-Dog, we ran a grid search with 16 configurations for regular pretraining, 16 configurations for PCGrad and 12 configurations for ATTITUD. More experimental details are available in Appendix C

# 6 RESULTS AND DISCUSSION

**Text Classification.** Table 1 shows the results for text classification. When the same data is used both for the auxiliary task of MLM and the primary classification task, TAPT and ATTITTUD both bring a similar improvement over Roberta (Imdb, Amazon columns). For the Cross-TAPT setting where different data is used for the auxiliary task and the primary task (Imdb + Amazon MLM, Amazon + Imdb MLM columns), TAPT does not perform as well as ATTITTUD. This highlights the advantage of ATTITTUD when the auxiliary task data distribution differ from the primary task distribution.

**Image Classification.** Our results are presented in Table 2. Both for MultiCifar100 (high resource setting) and Cifar10-Cat-vs-Dog (low resource setting), ATTITUD shows a strong improvement over baselines. In general, we find that primary-task aware pre-training (Multitasking, PCGrad, Ours) is better than vanilla pre-training which also performs better than having no pre-training at all. For MultiCifar100, we find that using $\eta_{aux} = (1.0, 1.0, -1.0), \eta_{prim} = 0.1$ worked best for 11 out of the 20 Cifar100 super-classes tasks. Note that $\eta_{aux} = (1.0, 1.0, -1.0)$ is an aggressive but novel configuration we introduce. Multitask learning and PCGrad produce better models on 6 and 3 tasks respectively. In the low-resource Cat-vs-Dog, setting ATTITUD produces a bigger boost in performance compared to baselines, with the best performing configuration being $\eta_{aux} = (1.0, 0.0, 0.0), \eta_{prim} = 0.01$. We posit that this configuration is successful because removal of the in-span components makes overfitting less likely. Applying the out-of-span components means the model learns features that do not harm the loss of the current mini-batch but could be useful later. Note that our best performing configurations are all novel and never an instantiation of PCGrad.

| Method | MultiCifar100 | Cifar10-Cat-vs-Dogs |
|---|---|---|
| No-Pretraining | 57.6 | $53.6 \pm 2.26$ |
| Vanilla Pre-training | 70.2 | $64.5 \pm 1.26$ |
| PCGrad | 75.6 | $64.2 \pm 1.10$ |
| Multitask | 75.5 | $65.3 \pm 1.35$ |
| Ours | **76.1** | **67.1±1.31** |

Table 2: Average Accuracy on MultiCifar100 and Cat-vs-Dog Cifar10 tasks. Cat-vs-Dog experiments are averaged over 5 runs

| Method | Average AUC Across 5 Pathologies |
|---|---|
| No-Pretraining | $78.3 \pm 0.87$ |
| Pretrained-ResNet | $81.4 \pm 1.34$ |
| Pretrained-ResNet + Ours | **83.3±0.71** |

Table 3: Results on ChexPert-5k task measured by average AUC (Area Under Roc-Curve). All experiments are averaged over 5 runs.

**Medical Imaging Transfer.** Table 3 shows our results on the ChexPert multi-label classification task. Per-pathology breakdowns are in Appendix C. Doing no pre-training at all performs worst.

Our method outperforms using a pre-trained Resnet18 model over Imagenet. We apply the end-task-aware ATTITUD over 100k ImageNet images after the initial pretraining and we reach 83.3% AUC, an improvement over 81.4%.

| Subspace | Canonical | Random | Unit_avg_grad | Randomized_SVD |
|---|---|---|---|---|
| Average Acc. | $51.42 \pm 2.09$ | $58.72 \pm 2.68$ | $59.13 \pm 2.08$ | **62.2±4.00** |

Table 4: Experiment conducted on Cat-vr-Dog Cifar10 dataset for different choices of subspace basis. We use $k = 5$ for Random and Randomized_SVD. This ablation uses a smaller hyper-parameter budget than Table 2

**Ablation Study.** Our approach relies on the top-k singular vectors from *randomized_svd* to define the basis to identify the positive and negative component of the auxiliary task gradient, see Section 4. This method is more accurate than several alternatives; see Table 4. Namely, we compare our choice to *random*, the basis spanned by $k$ randomly chosen orthogonal vectors in $\mathbb{R}^D$, *unit_avg_grad*, the basis spanned by the average primary task gradient, and *canonical*, the per-parameter basis. This ablation was performed under a more limited tuning budget (we cross-validated on configurations $(1, 1, 0)$ and $(1, 1, -1)$ only) than the full Cat-vs-Dog experiments from Table 2.

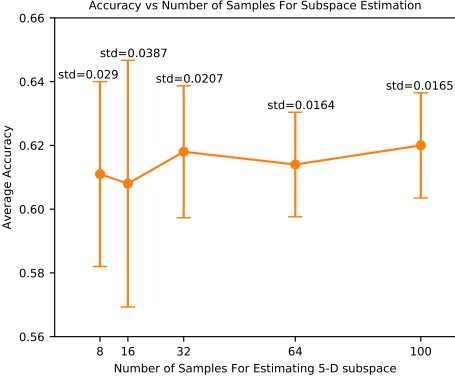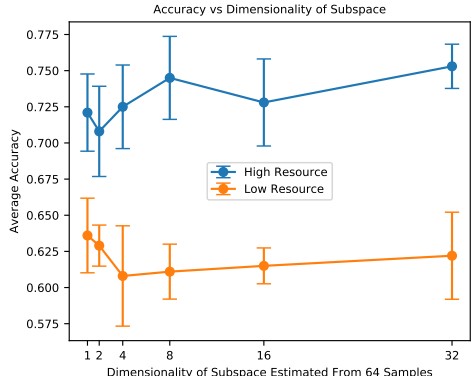

Figure 2: Averaged across 5 random initializations. **Left** We vary the number of samples used to estimate a 5-d subspace up to a maximimum of 100 (the total number of training examples in this low-resource setting). **Right.** We compare the effect of the dimensionality of the subspace in the low-resource (50 examples each for Cat, Dog classes) and high-resource (1000 examples each per class).

We also examine the number of samples to estimate the principal directions of the per-example primary task gradient. Larger sample sizes involve more computation but have limited benefit on average accuracy. Large sample sizes however reduce variance, as shown in Figure 2 (left). This is as expected since using more samples gives a higher fidelity estimate of the top-k singular vectors.

Another parameter of our algorithm is the size of our subspace, $k$. In general, we observe that in low-resource settings, it is better to operate on the auxiliary task gradient in a smaller dimensional subspace. The opposite holds for high-resource settings. This can be seen in Figure 2 (right). Whilst using a larger dimensional subspace captures a richer description of the $\boldsymbol{J}^*$, it also creates the risk of over-fitting especially in a limited data setting. This trade-off therefore has to be validated on a per-task basis.

# 7 CONCLUSIONS

In this work, we propose a new approach to training a model with additional help from an auxiliary task. Our method decomposes the gradients of the auxiliary task according to three directions, with positive, negative and neutral impact on the primary task. This decomposition allows a flexible re-weighting of the auxiliary task components and give rise to a family of training strategies,

which encompasses novel and existing approaches. We leverage insights from randomized linear algebra and automatic differentiation to scale the approach to large deep networks. Experiments in multitasking, pretraining and domain transfer over vision and text classification task demonstrate the empirical benefit of our framework.

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

# A    PROOF OF THEOREM 1

**Theorem.** *Let $\mathcal{L}^{\mathrm{aux}}(\theta_t)$ and $\mathcal{L}^{\mathrm{prim}}(\theta_t)$ represent the full batch losses of the auxiliary tasks and primary task respectively at step t. We assume the gradients of $\mathcal{L}^{\mathrm{aux}}$ and $\mathcal{L}^{\mathrm{prim}}$ are Lipschitz continuous with constant $L > 0$. Following the update rule : $\theta_{t+1} = \theta_t - \alpha \cdot \tilde{g}_{aux}$, where $\alpha \leq \frac{1}{L}$ is the learning rate, we are guaranteed :*

$$\mathcal{L}^{\mathrm{aux}}(\theta_{t+1}) \leq \mathcal{L}^{\mathrm{aux}}(\theta_t)$$
$$\mathcal{L}^{\mathrm{prim}}(\theta_{t+1}) \leq \mathcal{L}^{\mathrm{prim}}(\theta_t)$$

*If $\eta_- = 0$ and $\eta_\perp, \eta_+ \geq 0$*

*Proof.* Let $V_t \in \mathbb{R}^{K \times D}$ be the orthonormal matrix whose rows span the per-example primary task gradients $J^*$ at timestep $t$. The projections of the average primary task gradient $g_{prim} = \frac{1}{m} \sum_{i=1}^{m} J_{i,:}^*$ and average auxiliary task gradient $g_{aux}$ at iteration $t$ are

$$p_{prim} = V_t (g_{prim})^T$$
$$p_{aux} = V_t (g_{aux})^T$$

$p_{prim}$ and $p_{aux}$ will agree on some directions (same sign on those components). We use the operator $[x]_+$ to mark these directions of agreement. This operator preserves components that agree and sets those that disagree to zero. As an example given $p_{prim} = [1, 1, -1]$ and $p_{aux} = [1, 3, 10]$, $[p_{prim}]_+ = [1, 1, 0]$ and $[p_{aux}]_+ = [1, 3, 0]$. For directions that disagree (different signs of the respective components), we introduce the operator $[x]_-$. In the above example $[p_{prim}]_- = [0, 0, -1]$ and $[p_{aux}]_- = [0, 0, 10]$. Note that our operators are defined by comparing two vectors $x_1$ and $x_2$, Our operators have the following properties by definition :

$$x = [x]_- + [x]_+$$

and

$$[x]_+ \perp [x]_-, [x_1]_\pm \perp [x_2]_\mp$$

From Equation 2 :

$$\tilde{g}_{aux} = \eta_+ g_{aux}^+ + \eta_- g_{aux}^- + \eta_\perp g_{aux}^\perp$$

We can re-write this in terms of $[x]_\pm$ as :

$$\tilde{g}_{aux} = \eta_+ [p_{aux}]_+ + \eta_- [p_{aux}]_- + \eta_\perp (g_{aux} - p_{aux})$$

We now proceed to show the effect of the gradient descent update below on $\mathcal{L}^{\mathrm{aux}}(\theta_{t+1})$ and $\mathcal{L}^{\mathrm{prim}}(\theta_{t+1})$.

$$\theta_{t+1} = \theta_t - \alpha \cdot \tilde{g}_{aux} \tag{3}$$

How does this update affect the loss on the primary task loss $\mathcal{L}^{\mathrm{prim}}(\theta_{t+1})$?

$$\mathcal{L}^{\mathrm{prim}}(\theta_{t+1}) = \mathcal{L}^{\mathrm{aux}}(\theta_t - \alpha \cdot \tilde{g}_{aux})$$

$$\approx \mathcal{L}^{\mathrm{prim}}(\theta_t) - \alpha (\tilde{g}_{aux})^T g_{prim} \;\; \textit{(First order Taylor Expansion)}$$

$$= \mathcal{L}^{\mathrm{prim}}(\theta_t) - \alpha \left( \eta_+ [p_{aux}]_+ + \eta_- [p_{aux}]_- + \eta_\perp g_{aux}^\perp \right)^T g_{prim}$$

$$= \mathcal{L}^{\mathrm{prim}}(\theta_t) - \alpha \left( \eta_+ [p_{aux}]_+ + \eta_- [p_{aux}]_- + \eta_\perp g_{aux}^\perp \right)^T \left( [p_{prim}]_+ + [p_{prim}]_- \right)$$

$$= \mathcal{L}^{\mathrm{prim}}(\theta_t) - \alpha \left( \eta_+ \left( [p_{aux}]_+^T [p_{prim}]_+ + [p_{aux}]_+^T [p_{prim}]_- \right) + \eta_- \left( [p_{aux}]_-^T [p_{prim}]_+ + [p_{aux}]_-^T [p_{prim}]_- \right) \right)$$

$$= \mathcal{L}^{\mathrm{prim}}(\theta_t) - \alpha \left( \eta_+ [p_{aux}]_+^T [p_{prim}]_+ + \eta_- [p_{aux}]_-^T [p_{prim}]_- \right)$$

$$\leq \mathcal{L}^{\mathrm{prim}}(\theta_t) \;\; \textit{(if } \eta_- \leq 0, \; \eta_\perp, \eta_+ \geq 0 \textit{)}$$

Note that in going from line 3 to 4 in the proof above, we use the fact that $\left(\boldsymbol{g}_{aux}^{\perp}\right)^T \boldsymbol{g}_{prim} = 0$ since $\boldsymbol{g}_{aux}^{\perp}$ lies outside the subspace and $\boldsymbol{g}_{prim}$ lies inside it. For the last step of the proof, we use the observations below :

$$[\boldsymbol{p}_{aux}]_+[\boldsymbol{p}_{prim}]_+ \geq 0 \ \text{ since these directions agree in sign}$$
$$[\boldsymbol{p}_{aux}]_-[\boldsymbol{p}_{prim}]_- \leq 0 \ \text{ since these directions disagree in sign}$$
$$[\boldsymbol{p}_{aux}]_+[\boldsymbol{p}_{prim}]_- = 0 \ \text{ by the property of the } [x]_\pm \text{ operator}$$
$$[\boldsymbol{p}_{aux}]_-[\boldsymbol{p}_{prim}]_+ = 0 \ \text{ same motivation as above}$$

How does Equation 3 affect the auxiliary task loss $\mathcal{L}^{\mathrm{aux}}(\theta_{t+1})$?

$$\begin{aligned}
\mathcal{L}^{\mathrm{aux}}(\theta_{t+1}) &= \mathcal{L}^{\mathrm{aux}}(\theta_t - \alpha \cdot \tilde{\boldsymbol{g}}_{aux}) \\
&\approx \mathcal{L}^{\mathrm{aux}}(\theta_t) - \alpha\big(\tilde{\boldsymbol{g}}_{aux}\big)^T \boldsymbol{g}_{aux} \ \textit{(First order Taylor Expansion)} \\
&= \mathcal{L}^{\mathrm{aux}}(\theta_t) - \alpha\big(\eta_\perp \boldsymbol{g}_{aux}^\perp + \eta_+ \boldsymbol{g}_{aux}^+ + \eta_- \boldsymbol{g}_{aux}^-\big)^T\big(\boldsymbol{g}_{aux}^\perp + \boldsymbol{g}_{aux}^+ + \boldsymbol{g}_{aux}^-\big) \\
&= \mathcal{L}^{\mathrm{aux}}(\theta_t) - \alpha\big(\eta_\perp \|\boldsymbol{g}_{aux}^\perp\|^2 + \eta_+ \|\boldsymbol{g}_{aux}^+\|^2 + \eta_- \|\boldsymbol{g}_{aux}^-\|^2\big) \ \textit{(Cross terms cancel due to orthogonality)} \\
&\leq \mathcal{L}^{\mathrm{aux}}(\theta_t) \ \textit{(If } \eta_-, \eta_\perp, \eta_+ \geq 0)
\end{aligned}$$

Thus, choosing $\eta_- = 0$ ensures that we are minimizing both $\mathcal{L}^{\mathrm{aux}}(\theta_t)$ and $\mathcal{L}^{\mathrm{prim}}(\theta_t)$. We can combine this with the constraint on $\alpha \leq \frac{1}{L}$ to derive convergence guarantees after some $T$ steps as in optimization literature. $\qquad\square$

## B    RANDOMIZED MATRIX THEORY

---
**Algorithm 2:** `randomized_lowrank_approx` : Construct low rank approximation

---
**Require :** $\boldsymbol{J} \in \mathbb{R}^{m \times D}$ : Input Matrix
**Require :** $k$ : Rank of subspace
$\quad \Pi \sim \mathcal{N}(0, I) \in \mathbb{R}^{k \times m}$
$\quad \boldsymbol{C} = \Pi \boldsymbol{J}$
$\quad \boldsymbol{V} \leftarrow \mathrm{Gram\_Schmidt(C)}$
**Return :** $\boldsymbol{V} \in \mathbb{R}^{k \times D}$ : Low rank approximation of $\boldsymbol{J}$

---

The $\mathrm{Gram\_Schmidt}$ procedure orthogonalizes the rows of an input matrix.

## C    MORE EXPERIMENTAL DETAILS

**Image Classification** For MultiCifar100, unlike Rosenbaum et al. (2017); Yu et al. (2020) who use a 500-100 train-test split for examples under each fine-grained CIFAR 100 label, we include a validation set and therefore opt for a 400-100-100 train-validation-test split. We test on all 1000 test examples per class.

For Cat-vs-Dog, we use 100 examples from the training set as validation and test on all 1000 test examples per-class.

For Image Classification experiments, we perform pre-training with a learning rate of 1e-4 for all experiments and finetuning learning rate of 5e-4. These values were selected after coarse hyper-parameter search. In both pre-training and finetuning settings, we decay the learning rate by 0.5 if the validation loss has not improved over 4 epochs, up till a minimum learning rate of 1e-5. we use the Adam Optimizer (Kingma & Ba, 2014) with $\beta = (0.9, 0.999)$. We clip all gradient norms to 1.0 before performing gradient descent. We cross-validated dropout rates within the set $\{0.05, 0.1, 0.2, 0.3\}$ for both pre-training and finetuning steps. We cross validate $\eta_{prim}$ based on the relative sizes of primary and auxiliary task datasets. All experiments are averaged over 5 random seeds. For all our Vision experiments, we either recompute our subspace basis every $n = 5$ or $n = 10$ iterations. We find that $n$ is not as important as the other hyper-parameters, with the two choices showing similar

performance when the other hyper-parameters (learning rate and gradient norm clipping) are fixed to reasonable values.

Due to the fact that Yue et al (PCGrad) treat all tasks symmetrically, which is different from our primary-auxiliary setting, we introduced an extra parameter, $\alpha_{prim}$, for PCGrad to account for weighting the primary task. We cross validated values of $\alpha_{prim} \in \{0.1, 0.05, 0.01, 0.001\}$.

**Medical Imaging Transfer** Table 4 presents a more detailed breakdown of the ChexPert task. For 50k examples from Imagenet, our best performing configuration was $\eta_{aux} = (1.0, 0.0, -1.0)$. We did not use the primary task gradient directly for pre-training so $\eta_{prim} = 0.0$ for all cases. For ATTITUD, we use the same learning rates as in the Image classification setup above. For the No-Pretraining and Vanilla pretraining we cross-validated the learning rates for both finetuning and pre-training from the set $\{1e\text{-}3, 1e\text{-}4\}$. We cross-validated the same list of dropout values above.

| Method | No-Pretraining | Pretrain w Imgnet | Pretrained + Ours (50k) | Pretrained + Ours (100k) |
|---|---|---|---|---|
| Atelectasis | $76.0 \pm 1.82$ | $79.0 \pm 3.66$ | $\mathbf{81.6 \pm 1.38}$ | $\mathbf{81.8 \pm 0.80}$ |
| Cardiomegaly | $74.9 \pm 2.34$ | $75.8 \pm 4.04$ | $78.0 \pm 2.13$ | $\mathbf{80.7 \pm 1.79}$ |
| Consolidation | $83.2 \pm 2.26$ | $\mathbf{85.3 \pm 1.86}$ | $\mathbf{85.6 \pm 2.32}$ | $84.9 \pm 1.36$ |
| Edema | $79.5 \pm 1.27$ | $82.6 \pm 0.76$ | $\mathbf{85.2 \pm 1.23}$ | $84.7 \pm 1.78$ |
| P. Effusion | $77.9 \pm 1.88$ | $\mathbf{84.4 \pm 0.75}$ | $83.4 \pm 1.80$ | $\mathbf{84.3 \pm 0.65}$ |

Table 5: Results on ChexPert-5k tasks measured by average AUC (Area Under Roc-Curve)

**Text Classification** For our NLP experiments, we tried limiting the number of layers we applied ATTITUD to. We achieved good performance without applying ATTITUD to the word embedding layers (these were updated with untouched auxiliary task gradients). We cross-validated $\eta_{prim} = \{0.01, 0.05, 0.0025\}$. For all our NLP experiments, we either recompute our subspace basis every $n = 1$ or $n = 4$ times

For all experiments involving ATTITUD, We cross-validate the following choices of the subspace size $k \in \{5, 10, 20\}$ from $\boldsymbol{J}^* \in \mathbb{R}^{m \times D}$ using $m \in \{32, 64\}$. We recompute the subspace every 10 steps for vision experiments and every 4 steps for NLP experiments. We run all experiments for a maximum of 150 pretraining epochs and 500 finetuning epochs. We performed early stopping for all experiments if no improvement after 10 consecutive epochs.

**Ablation of Fraction of Norm within Subspace** The left pane of Figure 3 reinforces our intuition and confirms that our choice of the top-k singular vectors (*randomized_svd*) gives the best accuracy as averaged across 5 seeds. *random* is the basis spanned by $k$ randomly chosen orthogonal vectors in $\mathbb{R}^D$, *unit_avg_grad* is the basis spanned by the average primary task gradient whilst *canonical* uses the per-parameter basis. Note that $k = 5$ for *random* and *randomized_svd* whilst for *unit_avg_grad* and *canonical*, $k = 1$ and $k = D$ respectively. We use the fraction of the norm of sample gradients within a subspace as indicators of how *semantically* meaningful that choice of subspace is. We expect that a *semantically* meaningful choice of basis will achieve better generalization performance because it captures the essential parts of the gradient with $k \ll D$. *canonical* trivially captures all the norm of the sampled gradient vectors but because $k = D$, it generalizes poorly. Notice that only small fractions of the norms of sample primary and auxiliary task average gradients lie in the subspace for *random* and *unit_avg_grad*, whilst significant fractions lie in *randomized_svd*.

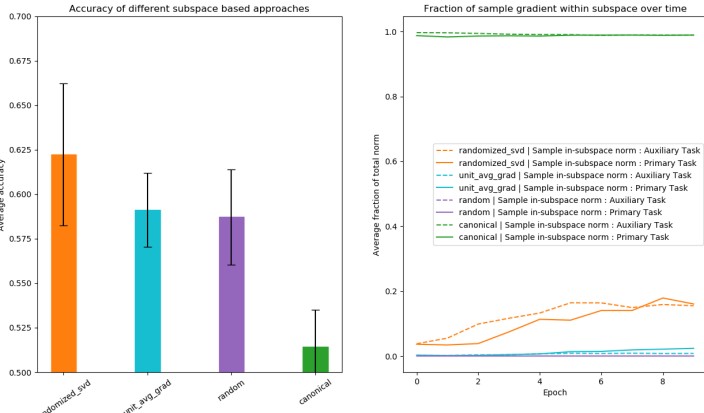

Figure 3: Experiment conducted on Cat-vr-Dog Cifar10 dataset. **Left** Averaged accuracy across 5 seeds of different choices of basis. Our choice, randomized_svd performs best. **Right** We look at the fraction of the norm of $g_{aux}$ within each subspace (dashed line). We also do so for a randomly sampled mini-batch of the primary task (solid line).

