# OpenReview forum: "AUXILIARY TASK UPDATE DECOMPOSITION: THE GOOD, THE BAD AND THE NEUTRAL"
_ICLR.cc/2021/Conference — ICLR 2021 Poster_

### Official Review · AnonReviewer2 · 2020-10-27
**Interesting idea, but questionable experimental setting**

**Rating:** 6
**Confidence:** 5

**Review:**

[Summary] This paper studies auxiliary learning, and propose a decomposition method on the auxiliary gradient into several components, and to select relevant/useful decomposed gradients to maximize the assistance of the primary task.

[Strength] How to adjust auxiliary tasks in a beneficial way is always a challenging problem in multi-task learning. The proposed solution on gradient decomposition is novel and interesting. The improved performance based on the proposed method seems to be non-trivial.

[Weakness] This paper however has some significant weaknesses which I will outline below.
-- Missing details. The subspace of the primary task gradient is composed of all training samples in the primary task, based on the definition of $\mathcal{S}$. However, for any reasonable-size training dataset which contains at least 10k training samples, this process seems to be extremely expensive to compute? The authors have introduced randomised approximation on decomposition step, but accumulate gradients seem to be already taking a lot of computation. I hope the authors could justify this.

-- Unfair experimental setting. My biggest concern is on hyper-parameter tuning on each component of the decomposed auxiliary task gradient. In Eq. 1, the authors decompose each auxiliary task gradient into three components: i) one is lying in the subspace of primary task gradient, ii) one is orthogonal to the primary task gradient ii) the final one is in conflict to the primary task gradient. By selecting different weightings on each component, the authors claim that this formulation can be degraded into prior auxiliary learning methods. I agree that this formulation is general. However, all of these component weightings are selected by hand, and seem to be very different across different datasets and tasks.

This gives a very unfair comparison to previous methods, simply because these methods are included in one of the hyper-parameter sets of corresponding weighting values. I have noticed that the authors have constrained the search space into 4 sets of weighting, but this does not justify this problem. Ideally, these weighting should be automatically computed during training, and varied based on the dataset.

In addition, the weighting for the primary task also varies across different datasets. So I am wondering whether these task weightings are consistent in baselines, and did authors perform a similar hyper-parameter search on baseline methods as well?

Other minor issues:
This formulation *cannot* be used as pre-training with only auxiliary gradients which are helpful to the primary task, since we do not know the primary task gradient beforehand. Even we know the primary task gradient, (0, 1, 0) is simply putting more weighting on the existing primary task gradient, so it's the same as to tune up the learning rate.

=======================

Rated up after authors' clarification.

---

> ### Author Response · Authors · 2020-11-17
> **Addressing AnonReviewer2 Concerns**
>
> We express our gratitude for providing feedback on our paper. We address your comments below.
>
> “This gives a very unfair comparison to previous methods, simply because these methods are included in one of the hyper-parameter sets of corresponding weighting values” -> This is inaccurate because we never report weight value configurations corresponding to baselines as our method in the paper. Note that for MultiCifar100 and Cat-vs-Dog, (where we compare against PCGrad) we report in Section 6 the best performing configurations as (1, 1, -1)  and (1, 0, 0) respectively. These configurations are novel under our formulation and so we are not simply reporting a setting of PCGrad as our optimal value. Sorry for the confusion, we have updated Section 5 to make this clearer.
>
> “The subspace of the primary task gradient is composed of all training samples in the primary task, based on the definition of S” - > We do not use all the available examples to estimate the subspace but rather a mini-batch of samples, since as you rightly pointed out, this would present a significant computational burden. We mention this in Section 4 under implementation “As stochastic optimization is prevalent in this setting, we construct subspace S from a mini-batch of primary task data”
>
> “Ideally, these weighting should be automatically computed during training, and varied based on the dataset.” ->  We agree that this would be the ideal setting. We decide to leave this as future work because automatic hyper-parameter selection is a challenging problem on its own. We however provide 4 hyper-parameter configurations that perform well across a reasonable set of tasks as guidance for practitioners.
>
> “This formulation cannot be used as pre-training with only auxiliary gradients which are helpful to the primary task, since we do not know the primary task gradient beforehand. Even we know the primary task gradient, (0, 1, 0) is simply putting more weighting on the existing primary task gradient, so it's the same as tuning up the learning rate” -> We would like to request further clarification on this point. Please note that our formulation relies on the fact that the primary task is known beforehand and thus cases where the primary task is unknown are outside the scope of our work

---

> > ### Comment · AnonReviewer2 · 2020-11-18
> > **Clarification**
> >
> > Thanks for your detailed feedback and comments. Since they are still some confusion here, let me further clarify my concerns.
> >
> > -- Fair Comparison to PCGrad. At the end of section 3, you have mentioned that PCGrad is one specific case under your general formulation, by setting $\eta_{aux} = (\alpha_{aux}, \alpha_{aux}, 0.0)$, which I agree. But when $\alpha_{aux} = 1.0$, PCGrad would be included as one of your 4 hyper-parameter sets in your evaluation, for which then of cause your method would out-perform it, since you can easily search a better weighting for $\alpha_{aux}$. I am not questioning the novelty of formulation. But since PCGrad is having a fixed set of auxiliary weightings, I think it would be unfair to compare it with non-fixed auxiliary weightings from your formulation.
> >
> > --  Further clarification to the last comment. We are not able to compute the gradient of the primary task, since the primary task is not used during auxiliary task pre-training. Even if you still compute the primary task gradient (not updating it by setting primary task weighting as zero), we only preserve the auxiliary gradient with the same descent direction from the primary task, then this is equivalent to just training the primary task as itself, with a different learning rate (1.0 in your case).

---

> > > ### Author Response · Authors · 2020-11-19
> > > **Response to clarification**
> > >
> > > “But since PCGrad is having a fixed set of auxiliary weightings, I think it would be unfair to compare it with non-fixed auxiliary weightings from your formulation.” - Thank you for your clarification. Please note that whilst it might appear that for PCGrad  $\alpha_{aux}$ is fixed, it actually is not. Given the relationship
> > >
> > > $(\alpha_{prim} \cdot g_{prim} +   \alpha_{aux} \cdot g_{aux}^{\perp} +  \alpha_{aux} \cdot g_{aux}^{+})$
> > >
> > > we can reduce this to :
> > >
> > > $\alpha_{aux}  \big(  (\frac{ \alpha_{prim}}{\alpha_{aux}} ) \cdot g_{prim} +   g_{aux}^{\perp} +  g_{aux}^{+} \big) $
> > >
> > > Note that re-writing this way reveals that we can have 1 degree of freedom - the ratio $\frac{\alpha_{prim} }{\alpha_{aux}}$ and we can massage $\alpha_{aux}$ into a scaling of the learning rate.  We note that in our Common Concerns section, we mention that we cross validated more values of $\alpha_{prim}$ for PCGrad : {0.1, 0.05, 0.01, 0.001} than for ATTITUD. Combining these parameters with other hyper-parameters means we actually give a higher hyper-parameter budget to PCGrad than ATTITUD overall. We mention this in Section 5 of the revised paper : “e.g. for Cifar10-Cat-vs-Dog, we ran a grid search with 16 configurations for regular pretraining, 16 configurations for PCGrad and 12 configurations for ATTITUD”
> > >
> > >
> > > “Further clarification to the last comment” - Thank you for the clarification on this comment. To show that we now understand you better, we restate your claim - The (0, 1, 0) configuration reduces to “ ... just training the primary task as itself with a different learning rate”. We note that this is not exactly correct. The configuration (0, 1, 0) is actually doing more than a simple weighting of the whole primary task gradient. It is upweighting only specific parts of the primary task gradients that agree with the auxiliary task. It only reduces to a simple upweighting of the whole primary task gradient if all the primary task gradient directions agree with that of the auxiliary task.  For example, assume our primary task decomposes to [1, 1, -1]. Naive upweighting the primary task gradient corresponds to a * [1, 1, -1] = [a, a, -a], where a > 1.0, which is equivalent to changing the learning rate. When we have an auxiliary task decomposition of [-1, 1, -1], this results in a final decomposition result of [0, 1, -1] under the \eta_{aux} choice of (0, 1, 0). Note that the first component has been completely removed, and not just weighed up. Thus, (0, 1, 0) isn’t just upweighting the whole primary task gradient, it is rather selecting the parts of the primary task gradient that agree with the auxiliary task gradient. This can be viewed as “reinforcing” directions that are mutually agreeable to both primary and auxiliary tasks - thus, this can prevent the traditional over-fitting associated with descent on the primary task alone (since we now only descend on directions with evidence from another task of being a 'good' direction of descent).

---

### Official Review · AnonReviewer3 · 2020-10-28
**A simple yet insightful idea is implemented while the experiments might not demonstrate the algorithm's full potential.**

**Rating:** 6
**Confidence:** 3

**Review:**

The work studies the auxiliary task selection in deep learning to resolve the burden of selecting relevant tasks for pre-training or the multitask learning. By decomposing the auxiliary updates, one can reweight separately the beneficial and harmful directions so that the net contribution to the update of the primary task is always positive. The efficient implementation is experimented in text classification, image classification, and medical imaging transfer tasks.

The first contribution is the decomposition algorithm and reweighting of the auxiliary updates. It is a simple idea with a nice insight of treating the primary task and the auxiliary tasks in different manners. The decomposition allows a reweighting on the updates to optimize the primary task as much as possible while keeping the auxiliary tasks providing improvable directions. The second contribution is an efficient mechanism to approximate and calculate the SVD of the Jacobian of the primary task. The mechanism is implemented from an existing randomized approximation method. The third contribution is a set of experiments verifying the proposed method. The experiments include text classification, image classification, and medical imaging transfer tasks. The most salient result is the 99% data efficiency to achieve improvable performance in the medical imaging transfer task.

Concerns

Besides the above positive contributions, following are some concerns from the observations:

1. The relative improvements comparing to the baselines in Table 1 and Table 2 do not seem as much as that in (Gururangan et al. 2020) and (Yu et al., 2020), respectively.

2. The weights reported in the experiments are 1 or -1 in the experiments. For instance, \eta_aux = (1, 1, -1) is reported in the image classification task.

The reader would expect much better improvements when given the freedom to reassign the weights on the decomposed directions, especially when the harmful part has a negated weight. Moreover, why are the values chosen in \eta 1 or -1? Would there be a nicer balance between, say, the beneficial and the harmful parts? For instance, would \eta = (1, 0.8, -0.9) be a better choice? It would be crucial that the authors can explain furthermore or support further experiments to confirm whether the potential of this decomposition algorithm is fully demonstrated or not.


=====================

Post Rebuttal

I have read the authors' response. All my concerns are addressed properly. However, I still doubt that even the corner cases of \eta have a better performance, would there be a systematic way to find the optimal parameters reflecting the true potential of this method. Thus, I will keep my score unchanged.

---

> ### Author Response · Authors · 2020-11-17
> **Addressing AnonReviewer3 Concerns**
>
> Thank you for your helpful feedback on our paper. We would like to address your concerns below.
>
> “The relative improvements comparing to the baselines in Table 1 and Table 2 do not seem as much as that in (Gururangan et al. 2020) and (Yu et al., 2020), respectively” -> Please see the Comparison to (Gururanga 2020) and (Yue 2020) under Common Concerns. We address this there but please let us know if you would like further clarification
>
> “The weights reported in the experiments are 1 or -1 in the experiments. For instance, \eta_aux = (1, 1, -1) is reported in the image classification task … For instance, would \eta = (1, 0.8, -0.9) be a better choice?” -> Please see the Hyperparameter Settings under Common Concerns, where we address this. We note that we are happy to include a more granular exploration of \eta_{aux} in the final version of the paper but as noted widening the search space would mean we allocate a much higher search budget to ATTITUD as opposed to the methods we compare to.  Please let us know if we can provide further clarification.

---

### Official Review · AnonReviewer4 · 2020-10-28
**Review on "Auxiliary task update decomposition: the good, the bad and the neutral"**

**Rating:** 5
**Confidence:** 2

**Review:**

##########################################################################

Summary:
Leveraging the power of the data-rich related tasks have been studied (e.g., pre-training and multitask learning). This paper points out that careful utilization of auxiliary task is required to gain enhanced performance in primary tasks. In order to prevent harming the performance of primary tasks, they suggest the method to decompose auxiliary updates into three directions which have positive, negative and neutral impact on the primary task.

##########################################################################

Reasons for score:

In this paper, it is highly interesting to see how to use a decomposition from the span of the primary task Jacobian to adapt auxiliary gradients and validate the proposed methodology on image and textual data. Even though this is an interesting setting and the technical solutions presented in the paper look reasonable, the idea seems to be pretty incremental as it stacks multiple existing techniques without many innovations.

##########################################################################

Pros:
1. The proposed methodology utilizes  automatic differentiation procedures and randomized singular value decomposition for efficient scalability.
2. The proposed framework allows the model to treat each auxiliary update independently by its impact on the task of interest, which seems to be interesting.

##########################################################################

Cons:
Authors need to perform more qualitative and quantitative analysis on the datasets to vilify the effectiveness of the proposed methodology.

##########################################################################

---

> ### Author Response · Authors · 2020-11-17
> **Addressing AnonReviewer4 Concerns**
>
> Thank you for your feedback. You mention that
> “Authors need to perform more qualitative and quantitative analysis on the datasets to vilify the effectiveness of the proposed methodology.” ->
> We have a hard time acting on this comment specifically. We would like to highlight that the experimental section already includes results testing how performance degrades in the scenario when the number of samples or the size of the subspace are very small. We also consider how that is affected by being in a low-resource and high-resource regime. We are happy to improve it. It would be helpful if you point at specific weaknesses or aspects we could address or improve.

---

### Official Review · AnonReviewer1 · 2020-11-01
**A good overall idea and well written, but the resulting algorithm might be too tedious to use in practice**

**Rating:** 6
**Confidence:** 3

**Review:**

Summary:
The authors present a general formulation of different settings in multitask learning (including pretraining regimes), in a setting where the goal is to get best performance for a pre-specified primary task and additional auxiliary tasks. The main idea is to divide the gradients on the auxiliary task into 2 subspaces: a subspace where the gradients influence performance of the primary task and a subspace where they only influence the auxiliary task without changing the loss on the primary task. Within the subspace that does have influence on the primary task, it is easy to compute directions that have a positive or negative effect on the primary task, which allows to create different learning schemes given the gradients that point toward: i) auxiliary influence only, ii) positive influence on auxiliary tass, iii) negative influence on primary task. Experimental results show improvements over previously identified meta learning methods on 2 natural language datasets and 3 image datasets.

Strengths:
The authors present a general framework for an important problem. It has many applications in a wide variety of fields and contributes to thinking about meta learning and pretraining in a more general way.
Explanations, illustrations and mathematical derivations are clear and easy to understand.
The authors show that with a careful choice of hyperparameters, their approach can improve performance, especially in settings with limited data on the primary task. The results on natural language datasets are also interesting, showing that they achieve a higher performance when the auxiliary task doesn’t exactly match the primary task data.

Weaknesses:
Some of the explanations and especially the proof fall apart, when considering the k-largest principal vector of J*. Since k<<D, the sum of all gradients in g_aux will clearly still have a big influence on the performance of the primary task.
The proposed algorithm introduces a lot of additional hyperparameters and not all of those hyper parameters are properly discussed. Eta_aux and eta_prim are properly discussed and the authors convincingly show that these parameters are implicitly set by other methods as well. Figure 2 shows an ablation study but does not help practitioners to set the discussed values, given the large variance over 5 runs.
The choice of the subspace for g_aux seems very critical and the provided experimental results and discussion are somewhat lacking. How can a random choice of subspace basis improve the results? Calculating the randomized_lowrank_approx is only done every n steps, but there is no mention of n later. Some experimental results should be provided to convince the reader that the basis does not change too much given 2 different batches from the primary task.

Other remarks:
The result in Table 2 is much better than the best result in Table 4 on the Cat-vs-Dogs experiment. What is the difference? Can the experiments in Table 4 be repeated to be more comparable to Table 2?
PCGrad most closely resembles this work. What type of subspace basis is used in that work? It would be interesting to see a direct comparison between PCGrad and the proposed method with eta_aux = (alpha_aux, alpha_aux,- alpha_aux) and using the same basis.
Early stopping after 10 epochs seems quite short and might explain some of the large variance in the results.

Minor remarks:
k is introduced without much explanation, which was a bit confusing on first reading. It should be clearly stated that it is a hyper parameter on first mention.

---

> ### Author Response · Authors · 2020-11-17
> **Addressing AnonReviewer1 Concerns**
>
> We appreciate the time and effort you took to provide us with feedback. We address your comments one by one below.
>
> “Since k<<D, the sum of all gradients in g_aux will clearly still have a big influence on the performance of the primary task”  ->
> The choice of a lower dimensional space (k) is a computational necessity and indeed the decomposition is not perfect. However, our strategy performs well despite its low dimension:
> figure 3 shows that with k = 5 and D = O(1M), we are able to capture up to 20% of the gradient norm. Thus, even though we don't capture all of the norm if k is small,  we capture a comparably large fraction with very few components.  With a large enough choice of k, but still with k << D, we can reduce the norm of the out-of-span component significantly so as to mitigate its influence on the primary task. Note that increasing k creates a tradeoff between computational efficiency and performance - we are encouraged that we observe significant improvement with relatively small values of k.
>
> “Not all of those hyper parameters are properly discussed”, “Calculating the randomized_lowrank_approx is only done every n steps but there is no mention of n later” -> We address this concern under the  Hyperparameter Settings section of Common Concerns. Please let us know if you would like any further clarification.
>
> “Figure 2 shows an ablation study but does not help practitioners to set the discussed values, given the large variance over 5 runs” ->  We attribute the higher variance as an unfortunate side effect of the limited data regime. The settings we explore, even when dubbed ‘high-resource’, involve training a relatively large model on much smaller datasets than typical. Moving to typical levels of data availability would distract from the low-resource setting we are most interested in.
>
> “Some experimental results should be provided to convince the reader that the basis does not change too much given 2 different batches from the primary task” -> We acknowledge that this would be a good ablation to include. We plan on including this experiment in the final version of the paper but we are still brainstorming an effective way to showcase this. This is especially  challenging because we use a randomized algorithm for computing the subspace, so we necessarily get a different subspace from batch to batch and measuring the difference between 2 subspaces is itself a non-trivial problem
>
> “How can a random choice of subspace basis improve the results” ->  For a ‘random’ basis, since very little of the gradient norm falls within the subspace, the auxiliary task gradient does not change much after gradient decomposition and this essentially reduces to normal end-task agnostic pre-training.
>
> “The result in Table 2 is much better than the best result in Table 4 on the Cat-vs-Dogs experiment. What is the difference?” -> For our ablation studies, we dedicated a limited computational budget for hyper-parameter selection for the methods explored in Table 4. Table 2 reflects a more expanded set of hyper-parameter configurations to reflect the strength of the baselines being compared to. Note that the baselines in Table 2 also enjoy expanded resources for hyper-parameter search
>
> “Early stopping after 10 epochs seems quite short and might explain some of the large variance in the results.” -> We note that we do not early stop after just 10 epochs. Rather, we early stop if there has been no improvement on the end-task validation loss after 10 epochs. We run all experiments for a maximum of 150 pre-training epochs and 500 fine-tuning epochs. We updated the appendix to make this clearer.

---

### Author Response · Authors · 2020-11-17
**Common Concerns**

We would like to thank all reviewers for their thoughtful comments and for spending time to provide feedback on our work. We address some common concerns amongst  the reviews here.

[General Correction]

We would like to submit the following correction to our paper.
For Table 2, our model is actually initialized from the Imagenet pretrained ResNet model and not from scratch. We have updated this table to reflect this fact. Thus, to further clarify, row 2 in Table 2 is the result obtained from finetuning a pre-trained ResNet-18 directly on the ChexPert-5k dataset. For rows 3 of Table 2, we start with a pre-trained ResNet-18 and finetune with further Imagenet data using ATTITUD before final direct training with ChexPert-5k.
We  sincerely apologize for any confusion. We have updated Table 2 and the text describing the result to reflect this fact.



[Comparison to (Gururanga 2020) and (Yue 2020)]

For our text classification experiments, since we are directly comparing to (Gururangan et al 2020), we used exactly the same hyper-parameters they used to train their classification heads. Our hyper-parameter optimization diverges from theirs when considering the pre-training done on BERT before final classification. Here, we downloaded their (already extensively cross-validated) pretrained TAPT models to compare against our model whose hyper-parameter settings are provided in Appendix C. We note that our reported results are slightly higher, ~0.5%, than those presented in (Gururangan et al 2020) but the gaps between methods are consistent. We attribute this to using a slightly different environment with a newer version of pytorch and the huggingface library.

Regarding Yue 2020, we note that the following differences between our experimental setup and theirs as noted in Section 5 and Appendix C respectively
We use a different architecture than used in (Yue 2020) and (Rosenbaum  2017) because that architecture can only be trained using Reinforcement-Learning. We use classical optimization for training with ATTITUD and hence defaulted to a simpler architecture amenable to this type of optimization.
Whilst (Yue 2020) and (Rosenbaum  2017) use a 500-100 train test split, we use a 400-100-100 train-val-test split. We thus train on less data
Since we are operating in the asymmetrical task setting, we train a different model for each of the 20 tasks, treating the remaining 19 tasks as auxiliary tasks. Yue 2020 train a single model (specialized architecture and different training regime from ours) to perform all 20 tasks at once



[Hyperparameter Settings]

AnonReviewer2 expresses concern that our hyper-parameter settings are unfair to the methods we compared with.
As already mentioned above under the comparison section, for our text classification experiments, we use the same set of hyper-parameters used by Gururangan et al. Gururangan et al make available extensively cross-validated TAPT models which we compare against our method thus we believe this presents a fair comparison.
For Yue et al (PCGrad), note that they treat all tasks symmetrically which is different from our primary-auxiliary task setting. We thus introduced \alpha_{prim} as an extra hyper-parameter for PCGrad to reflect this asymmetry. We cross validated values of  \alpha_{prim} in {0.1, 0.05, 0.01, 0.001} .  We also cross-validated dropout values as mentioned in Appendix C. We apologize for not making this clearer in the paper and have updated the appendix to reflect this.

Both AnonReviewer2 and AnonReviewer3 express concern over the fact that we explore 4 configurations for our auxiliary task parameters \eta_{aux}. We explored other configurations, though not extensively, but decided to highlight these 4 because they are more interpretable and thus easily justifiable by practitioners. Note that these 4 configurations were further cross validated via grid search with different values of \eta_{prim} and dropout, with the optimal values discovered discussed in Section 6. We view the fact that we are able to get performance improvements by restricting ourselves to the corners of the search space as an indicator of the strength of our method.

AnonReviewer1 raises concern about the lack of information about, n, how frequently we recompute the basis. For our Vision experiments, we cross-validate n = {5, 10}, and for Text classification, we explore n = {1, 4}. As we mention in Section 4, we apply gradient clipping and use a relatively small learning rate such that infrequently computing the basis does not degrade performance. We have updated Appendix to reflect this and we will include more extensive exploration of the impact of varying n in the final paper.

---

### Decision · Program_Chairs · 2021-01-07
**Final Decision**

**Decision:**

Accept (Poster)

**Comment:**

After engaging in some good interactive discussions all but one reviewer settled on a rating of marginal accept. The most negative reviewer didn't really provide a clear enough explanation of what was lacking in the work. The other reviewers felt that the observed gains for this multi-task learning framework were clear enough that the work is worthy of some attention by the community. The AC recommends acceptance, but one may consider this recommendation as a just past the line for acceptance recommendation.